# CT and MR utilization and morbidity metrics across Body Mass Index

**Mina Dawod**[1], **Paul Nagib**[1], **John Zaki**[1], **Luciano M. Prevedello**[2], **Amna A. Ajam**[2], **Xuan V. Nguyen**[2]*

1 The Ohio State University College of Medicine, Columbus, OH, United States of America, 2 Department of Radiology, The Ohio State University College of Medicine, Columbus, OH, United States of America

* xuan.nguyen@osumc.edu

**Data Availability Statement:** The manuscript reports patient data that are accessible to researchers only as aggregate results of structured data queries. All aggregate data relevant to this manuscript that are available to the authors have

## Abstract

### Objective

Obesity is a high-morbidity chronic condition and risk factor for multiple diseases that necessitate imaging. This study assesses the relationship between BMI and same-year utilization of CT and MR imaging in a large healthcare population.

### Methods

In this retrospective population-based study, all patients aged ≥18 years with a documented BMI in the multi-institutional Cosmos database were included. Cohorts were identified based on ≥1 documented BMI in 2021 within pre-defined ranges. For each cohort, we assessed the percentage of patients undergoing head, neck, chest, spine, or abdomen/pelvis CT and MR during the same year. Disease severity was quantified based on emergency department (ED) visits and mortality.

### Results

In our population of 49.6 million patients, same-year CT and MR utilization was 14.5 ±0.01% and 6.0±0.01%, respectively. The underweight cohort had the highest CT (25.8±0.1%) and MR (8.01 ± 0.05) imaging utilization. At high extremes of BMI (>50 kg/m$^2$), CT utilization mildly increased (18.4±0.1%), but MR utilization decreased (5.3±0.04%). While morbidity differences may explain some BMI-utilization relationships, lower MR utilization in the BMI>50 cohort contrasts with higher age-adjusted mortality (1.8±0.03%) and ED utilization (32.4 ±0.1%) in this cohort relative to normal weight (1.5±0.01% and 25.7±0.02%, respectively).

### Conclusion

Underweight patients had disproportionately high CT/MR utilization, and high extremes of BMI are associated with mildly higher CT and lower MR utilization than the normal weight cohort. The elevated mortality and ED utilization in severely obese patients contrasts with their lower MR imaging utilization. Our findings may assist public health efforts to accommodate obesity trends.

been uploaded as supplementary material. Patient-level source data records are owned by a third party (Epic Systems, Verona, WI) and are not individually accessible by the authors. Access to the Epic Cosmos platform may be requested at https://cosmos.epic.com/#request-access.

**Funding:** The author(s) received no specific funding for this work.

# 1. Introduction

Obesity is a high-morbidity chronic disease with a prevalence of ~42.5% [1], and achieving or maintaining optimal weight is often challenging [2]. Suboptimal Body Mass Index (BMI) may contribute to or complicate various disease processes, including stroke, heart disease, other cardiovascular disease, metabolic diseases, arthritis, and some cancers [3, 4]. While the potential impact of obesity trends on healthcare expenditures has been recognized, understanding precisely how BMI affects population healthcare utilization can be challenging due to a complex interaction of multiple health-related and socioeconomic variables that may vary across geographic regions and populations.

Obesity-related healthcare costs in the U.S. in 2021 totaled around 173 billion dollars, with annual ranges between 147 to 210 billion dollars [5–7]. Comparatively, stroke and heart disease combined across all patient populations cost the U.S. health care system 216 billion dollars per year [8]. Medical imaging, particularly relatively high-priced imaging modalities like computed tomography (CT) and magnetic resonance (MR), is one area of healthcare spending that has been targeted for cost reduction [9]. Given the rising incidence of BMI in the general population, it is reasonable that downstream consumption of imaging scans for obesity-related disorders would also rise, complicating national attempts to reduce healthcare expenditures [10, 11]. However, the relationship between BMI and radiologic imaging has not been studied extensively [11]. In addition, many existing studies on obesity-related healthcare expenditures rely on claims data from specific payors or from institutions in specific geographic areas. Single-institution studies may be limited due to geographic or population biases, while Medicare claims databases, although large, lack age diversity. Consequently, there is a need for recent data on imaging utilization among large, geographically diverse, multi-payor patient populations.

Given the need to better characterize the impact of BMI on imaging utilization at the population level, we aim to quantify, in a large multi-institutional dataset of existing electronic medical records, the association between BMI recorded in a given year and utilization of CT or MR imaging in the same year. We hypothesize that population-level metrics of CT and MR imaging vary across BMI ranges. A secondary aim of the study is to examine the above associations in relation to measures of morbidity and mortality and health care access.

# 2. Materials and methods

This retrospective cohort study utilizes existing data that cannot be linked to individual human subjects. The study design was reviewed by the Office of Responsible Research Practices at the Ohio State University and was deemed to be exempt under human subjects research guidelines (study # 2016E0244). Aggregate counts were obtained of adult patients meeting pre-specified data criteria using Cosmos (Epic Systems, Verona, WI), an informatics-based application that provides a data framework to aggregate de-identified and de-duplicated electronic health data for research [12]. At the time of project initiation, this HIPAA-limited data set consisted of more than 149 million patients from 142 different organizations spanning all 50 U.S. states. We used the Population data model within this application to obtain aggregate numbers of patients with BMI under the Vitals group of data filters. The authors did not have access to data that could identify individual participants during or after data collection. The data for this study were accessed on 4/26/2024. Patients under 18 years of age or lacking a BMI measurement in 2021 were excluded.

For the 2021 calendar year, queries were performed to retrospectively identify the number of patients with BMI recorded (kg/m$^2$) in that calendar year that falls into one of the National Heart, Lung, and Body Institute (NHLBI) ranges of <18.5 (underweight), 18.5–24.9 (normal),

25–29.9 (overweight), 30–34.9 (Obese Class 1), 35–39.9 (Obese Class 2), ≥40 (Obese Class 3) [13]. In addition, because feasibility of CT and MR imaging may be affected at high extremes of BMI, and the Cosmos dataset was sufficiently large to permit further stratification, we subdivided the Class 3 group into a 40–40.9 cohort and a >50 cohort in this analysis. In each BMI cohort, the number of the subset of patients who had an imaging exam of interest in the same calendar year was recorded. For this study, the billed imaging exams of interest were any head, soft tissue neck, chest, abdomen/pelvis, or spine CT or MRI exams. To simplify analysis, no CT or MR angiograms were included. MR and CT utilization was recorded separately for each body region, but for most of the data analysis, grouped CT or grouped MR utilization variables were used, denoting counts of patients who received any CT or MR exam of interest in the 2021 calendar year. Note that utilization in this analysis is treated as a binary patient variable, such that the counting method does not differentiate between a patient undergoing multiple imaging exams of interest and a patient undergoing only one such exam in the same calendar year.

To explore potential confounding demographic effects, CT and MR utilization was also examined after further sub-stratifying the population by age (18–39 years, 40–65 years, or >65 years old), by sex (male or female), and by quartile of CDC Social Vulnerability Index socioeconomic status (SES) ranking based on patients' most recent ZIP code of residence and census tract data. In this ranking system, a lower vulnerabililty index represents more favorable socioeconomic conditions. Data are presented in the form of quartiles of social vulnerability as follows, in increasing order of vulnerability: (socioeconomic class quartile 1: <25th percentile % (SE C1), socioeconomic quartile class 2: 25-50th percentile % (SE C2), socioeconomic quartile class 3: 50–75%th percentile (SE C3), and socioeconomic quartile class 4: 75-100th percentile % (SE C4). Patients without a recorded SES were included in the total study population but were excluded from SES-stratified data analysis. To further assist in interpretation of utilization data, similar analyses were performed using additional outcome variables. Presence of any radiology department encounter in 2021 was examined to permit comparison of MR and CT utilization to radiology utilization in general, and presence of an emergency room (ER) encounter in 2021 and documentation of deceased status in 2021 were used to characterize morbidity and mortality risks of each cohort. Because mortality is strongly age-dependent, mortality data for each BMI cohort are reported as age-adjusted mortalities using proportions of each age group within the entire study population to appropriately weight age-specific values within each BMI cohort.

To facilitate interpretation in the setting of population heterogeneity, we performed a subgroup analysis examining the relationship between CT/MR imaging utilization and BMI among subcohorts defined by having any encounter within the study period associated with the following diagnosis groupings: Essential (primary) hypertension (ICD-10-CM: I10), Ischemic heart disease (I20.*—I25.*), Heart failure(ICD-10-CM: I50.*), Disorders of lipoprotein metabolism and other lipidemias (ICD-10-CM: E78.*), Type 1 diabetes mellitus (ICD-10-CM: E10.*), Type 2 diabetes mellitus (ICD-10-CM: E11.*), Cerebrovascular disease (ICD I60-I69), Back pain/Dorsalgia (ICD-10-CM: M54), and Cancer.

## 3. Data analysis

From the aggregate counts described above, for each cohort or subgroup defined by BMI and specified demographic or encounter criteria, the counts of the subset of the cohort meeting the outcome variable of interest (e.g., imaging utilization) were calculated to yield a per-population percentage. Utilization data are presented for each BMI range as percentages of patients undergoing CT or MR imaging in the same calendar year, with imaging by anatomical region

also separately reported. For outcome data in the form of proportions, 95% confidence intervals were computed based on the normal approximation of the binomial. Inferential statistics were performed on the main imaging utilization metrics for selected BMI cohorts relative to the relevant normal BMI cohort. This was performed as a two-tailed two-sample Z-test, and Z statistics with absolute value > 1.96 are considered statistically significant at an alpha of 0.05 and Z statistics with absolute value > 3.89 are considered statistically significant at an alpha of 0.0001.

## 4. Results

### 4.1 Study population

Our population of all patients aged 18 years or older with a documented BMI in the Cosmos database in 2021 consisted of 49.6 million patients. Table 1 lists the size of each BMI cohort included in the study. The largest BMI cohort is this population was the overweight class [N = 17.8 million, (35.8%)], followed by the normal-weight cohort [N = 14.6 million, (29.4%)]. The sizes of subgroups stratified by age and SES are also shown in Table 1.

### 4.2 Estimating effects of BMI on CT and MR utilization

Per-capita utilization of the study population was $14.5 \pm 0.01\%$ and $6.0 \pm 0.01\%$ for CT and MR, respectively. Across the examined BMI ranges, CT and MR utilization varied nonlinearly (Fig 1A and 1B). The underweight cohort was observed to have disproportionately higher CT and MR utilization ($25.8 \pm 0.1\%$ and $8.01 \pm 0.05\%$, respectively) relative to the normal BMI cohort (p<0.0001). Normal-weight, overweight, and mildly or moderately obese cohorts had CT and MR utilization of 16–17% and 5.9–6.4%, respectively. At the higher end of the BMI spectrum (BMI > 50 kg/m²), CT utilization increased to $18.4 \pm 0.1\%$ (p<0.0001), and MR utilization decreased to $5.3 \pm 0.04\%$ (p<0.0001).

Imaging utilization was further stratified by body region to assess whether the observed relationships between obesity and imaging utilization are affected by imaged anatomy (Fig 1C and 1D). Most performed CT exams were of the abdominopelvic region, followed by head CT and chest CT, and most performed MR exams were head MRs, followed by spine and abdomen/pelvis MRs. For abdominopelvic imaging, increases in BMI above the overweight class are associated with higher CT utilization but lower MR utilization. For head CTs, spine CTs, and chest CTs, increasing BMI is associated with decreasing utilization except at the high extremes of BMI (>50 kg/m²). Head MRs showed monotonic decrease in utilization with

**Table 1. Number of patients (in millions) in each BMI cohort included in the study population (Total), stratified by age, sex, and socioeconomic status (SES) vulnerability quartile.** Entries are expressed in millions of patients (e.g., 0.57 = 570,000 patients).

|  | <18.5 | 18.5–24.9 | 25–29.9 | 30–34.9 | 35–39.9 | 40–50 | >50 | Total |
|---|---|---|---|---|---|---|---|---|
| Age 18–39 years | 0.57 | 5.53 | 4.43 | 2.89 | 1.69 | 1.22 | 0.32 | 14.50 |
| Age 40–65 years | 0.26 | 4.52 | 7.02 | 5.65 | 3.24 | 2.15 | 0.54 | 19.90 |
| Age > 65 years | 0.41 | 4.54 | 6.31 | 4.18 | 1.94 | 0.98 | 0.16 | 15.24 |
| Female | 0.84 | 9.06 | 9.27 | 7.00 | 4.23 | 2.97 | 0.73 | 28.67 |
| Male | 0.41 | 5.53 | 8.49 | 5.72 | 2.64 | 1.38 | 0.29 | 20.97 |
| SES Vulnerability Quartile 1 | 0.33 | 4.43 | 5.00 | 3.18 | 1.55 | 0.88 | 0.18 | 13.41 |
| SES Vulnerability Quartile 2 | 0.27 | 3.30 | 4.03 | 2.86 | 1.52 | 0.94 | 0.21 | 11.17 |
| SES Vulnerability Quartile 3 | 0.28 | 3.15 | 3.99 | 3.00 | 1.68 | 1.09 | 0.26 | 11.31 |
| SES Vulnerability Quartile 4 | 0.35 | 3.57 | 4.57 | 3.57 | 2.07 | 1.41 | 0.37 | 13.30 |
| Total | 1.25 | 14.60 | 17.76 | 12.72 | 6.87 | 4.35 | 1.02 | 49.64 |

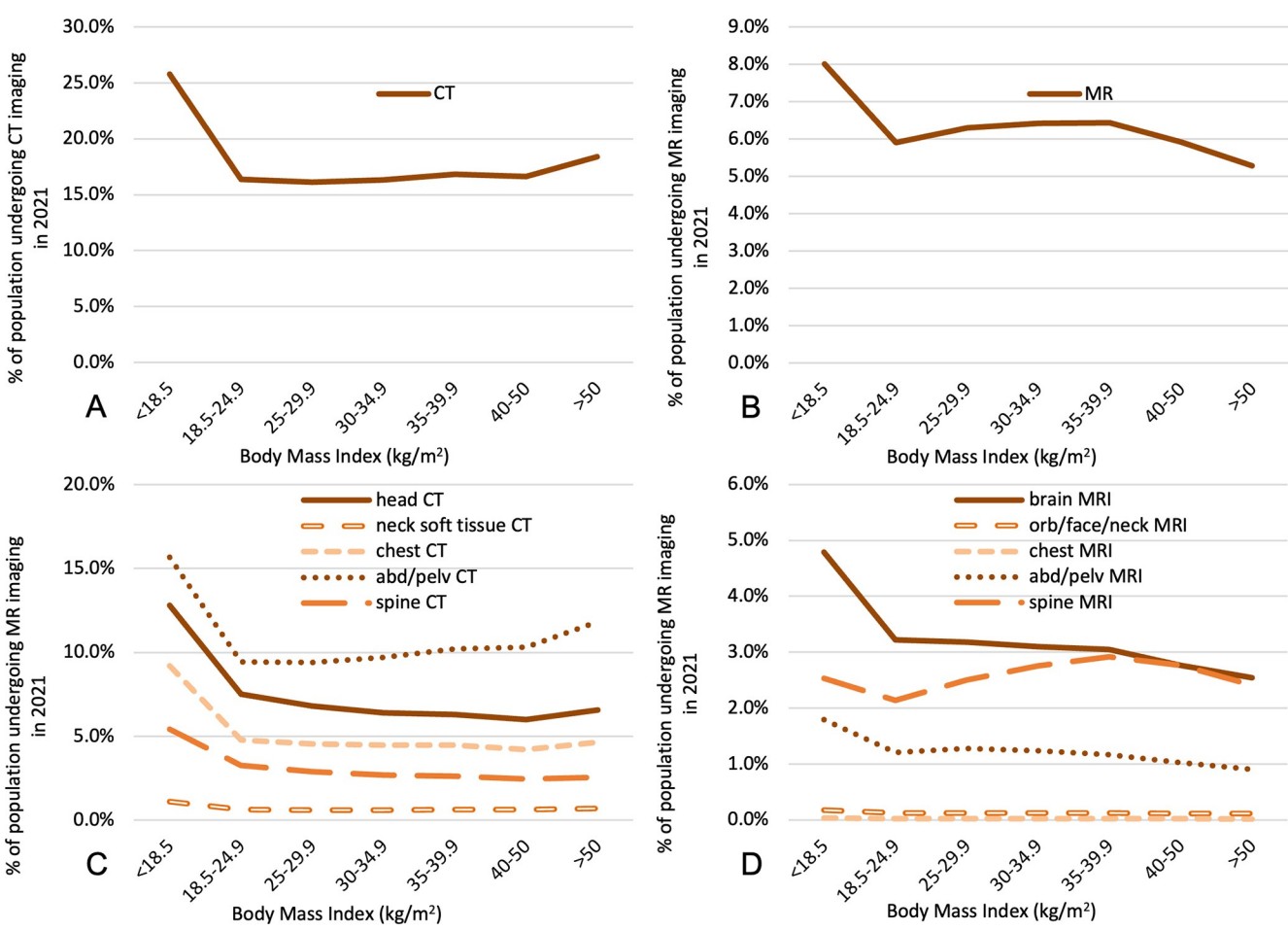

**Fig 1.** Association between BMI and the percentage of the cohort undergoing any CT (**A**), any MR (**B**), body region-specific CT (**C**), and region-specific MR (**D**) in 2021.

increasing BMI class. Spine MR utilization increased with increasing BMI from normal to the moderately obese range, beyond which it declined. The less common imaging exams (neck CT, neck MR, chest MR, and abdominopelvic MR) showed a subtle negative association between utilization and BMI across most of the BMI range examined.

To explore potential confounding effects of age, sex, and health care access on imaging utilization, we computed the above BMI-utilization relationships after subdividing the population by age, sex, or socioeconomic status quartile. Increasing age is associated with increased CT and MR utilization for all BMI groups. When partitioning the population into 3 broad age groups, the disproportionately high utilization observed among underweight adult patients of all ages persists for the two older age groups (Fig 2A). However, the higher utilization among the underweight group is less apparent for the youngest age group examined (ages 18–39), for whom underweight status confers CT and MR utilization of 12 ± 0.1% and 3.3 ± 0.05% (p<0.0001 relative to normal BMI), respectively, similar to CT and MR utilization rates of 14 ± 0.1% and 3.4 ± 0.06% (p<0.0001 relative to normal BMI), respectively, for the BMI > 50 group. For comparison, the normal-weight groups for this age range shows CT and MR utilization of 9.2 ± 0.02% and 2.5 ± 0.01%, respectively. The decrease in MR utilization observed with higher BMI in the study population is predominantly due to trends in the older age groups (Fig 2B), which have disproportionately higher impact on utilization rates. When

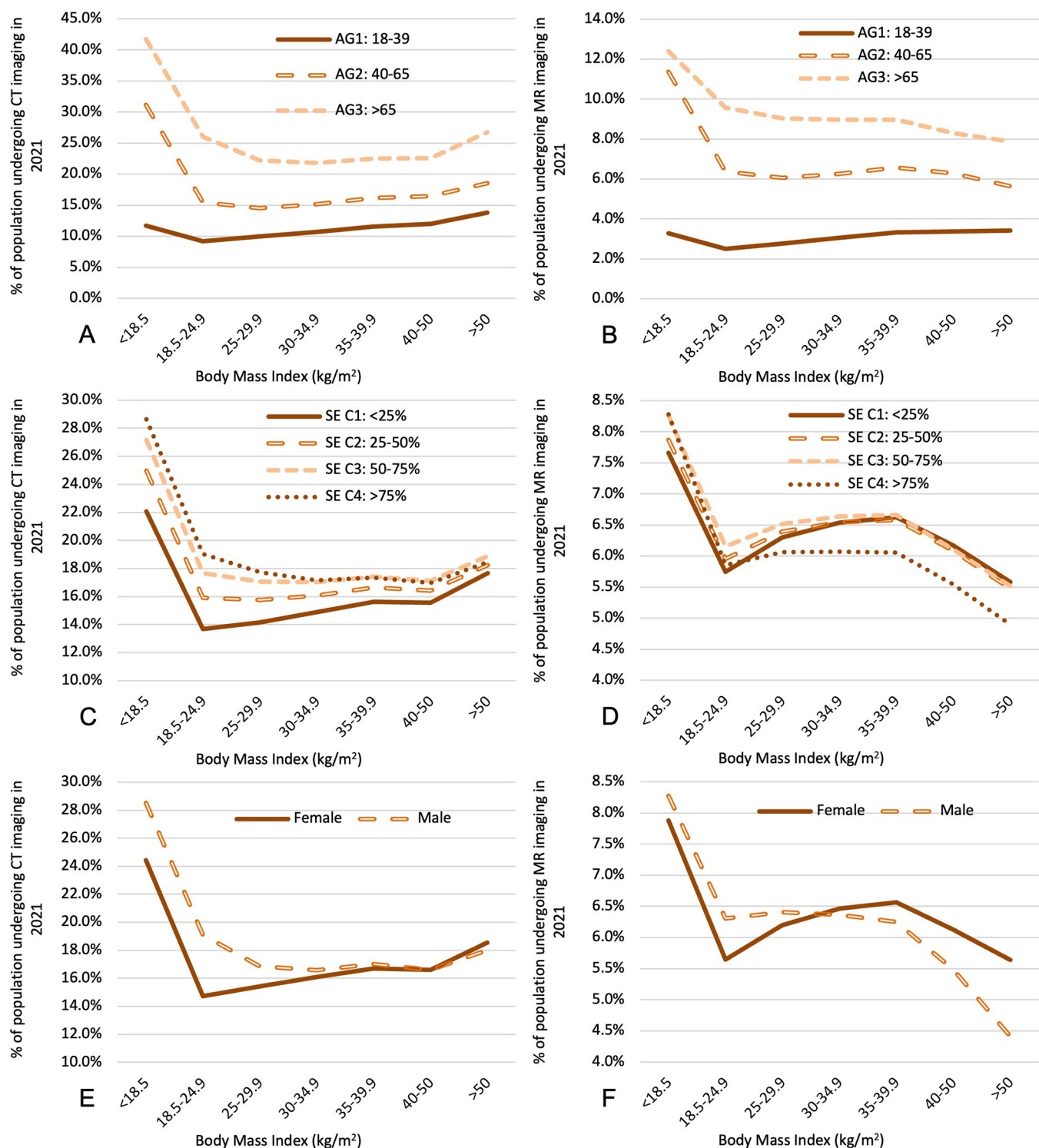

**Fig 2.** Association between BMI and the percentage of the cohort undergoing CT imaging in 2021 stratified by age (**A**), SES (**C**), and sex (**E**) and MR imaging in 2021 stratified by age (**B**), SES (**D**), and sex (**F**). AG = age group; SE = socioeconomic status social vulnerability category (C1 presents the lowest quartile of vulnerability).

**Table 2. CT utilization rates for each combination of sex, age (in years), SES SVI vulnerability quartile, and BMI.**

| Sex | Age | SES SVI | <18.5 | 18.5–24.9 | 25–29.9 | BMI 30–34.9 | 35–39.9 | 40–50 | >50 | Total |
|---|---|---|---|---|---|---|---|---|---|---|
| F | 18–39 | Q1 | 8.7% | 6.4% | 7.3% | 8.6% | 10.0% | 10.9% | 13.3% | 6.9% |
| F | 18–39 | Q2 | 10.7% | 7.9% | 8.9% | 10.0% | 11.3% | 12.0% | 14.2% | 8.4% |
| F | 18–39 | Q3 | 11.9% | 9.1% | 9.9% | 10.9% | 12.1% | 12.5% | 14.7% | 9.4% |
| F | 18–39 | Q4 | 12.8% | 9.9% | 10.4% | 11.0% | 11.8% | 12.2% | 14.0% | 9.8% |
| F | 18–39 | N/A | 10.2% | 8.2% | 10.1% | 10.7% | 11.9% | 12.4% | 13.6% | 8.9% |
| F | 40–65 | Q1 | 20.0% | 9.9% | 11.2% | 12.9% | 14.1% | 14.6% | 17.0% | 10.5% |
| F | 40–65 | Q2 | 25.0% | 12.4% | 13.3% | 14.7% | 15.8% | 16.2% | 18.3% | 12.5% |
| F | 40–65 | Q3 | 29.2% | 15.0% | 15.2% | 16.3% | 17.1% | 17.2% | 19.4% | 14.1% |
| F | 40–65 | Q4 | 32.6% | 17.6% | 16.6% | 16.8% | 17.6% | 17.7% | 19.3% | 15.2% |
| F | 40–65 | N/A | 24.0% | 11.7% | 13.0% | 14.4% | 15.2% | 15.5% | 17.9% | 12.0% |
| F | >65 | Q1 | 36.6% | 22.5% | 21.1% | 21.4% | 22.0% | 22.1% | 26.2% | 19.7% |
| F | >65 | Q2 | 37.8% | 23.7% | 21.8% | 21.8% | 22.5% | 22.4% | 25.9% | 20.3% |
| F | >65 | Q3 | 39.7% | 25.3% | 22.8% | 22.6% | 23.2% | 23.2% | 27.0% | 21.2% |
| F | >65 | Q4 | 40.8% | 27.0% | 23.6% | 22.6% | 23.1% | 23.1% | 27.0% | 21.7% |
| F | >65 | N/A | 38.7% | 23.8% | 20.5% | 19.8% | 20.8% | 20.6% | 28.0% | 19.4% |
| M | 18–39 | Q1 | 9.5% | 7.9% | 8.5% | 9.3% | 10.0% | 10.3% | 12.1% | 7.8% |
| M | 18–39 | Q2 | 11.9% | 10.0% | 10.3% | 10.7% | 11.2% | 11.3% | 12.3% | 9.5% |
| M | 18–39 | Q3 | 14.0% | 11.6% | 11.8% | 12.0% | 12.3% | 12.4% | 13.0% | 10.8% |
| M | 18–39 | Q4 | 15.7% | 13.3% | 13.4% | 13.1% | 13.0% | 12.8% | 13.7% | 12.1% |
| M | 18–39 | N/A | 13.4% | 12.5% | 12.6% | 13.1% | 13.1% | 13.3% | 12.9% | 11.6% |
| M | 40–65 | Q1 | 37.9% | 14.2% | 11.6% | 12.4% | 13.6% | 14.2% | 16.3% | 11.2% |
| M | 40–65 | Q2 | 40.8% | 18.5% | 14.2% | 14.4% | 15.3% | 15.5% | 17.7% | 13.4% |
| M | 40–65 | Q3 | 42.2% | 21.7% | 16.5% | 15.9% | 16.6% | 16.8% | 18.5% | 15.2% |
| M | 40–65 | Q4 | 44.4% | 24.8% | 18.7% | 17.4% | 17.9% | 17.6% | 19.3% | 17.1% |
| M | 40–65 | N/A | 38.2% | 21.1% | 15.3% | 14.6% | 15.5% | 16.7% | 19.5% | 14.5% |
| M | >65 | Q1 | 50.4% | 26.5% | 20.8% | 20.6% | 21.6% | 21.7% | 26.3% | 19.7% |
| M | >65 | Q2 | 50.0% | 28.0% | 21.9% | 21.3% | 22.2% | 22.2% | 26.8% | 20.6% |
| M | >65 | Q3 | 50.5% | 29.5% | 22.9% | 21.9% | 22.5% | 22.7% | 27.5% | 21.5% |
| M | >65 | Q4 | 50.4% | 30.9% | 23.7% | 22.2% | 22.7% | 22.5% | 27.9% | 22.5% |
| M | >65 | N/A | 48.3% | 26.1% | 19.4% | 18.4% | 20.1% | 21.7% | 22.7% | 18.7% |

examining only patients 18–39 years, overweight and obese patients show higher MR utilization than the normal-weight group. SES has a subtle effect on overall utilization rates (Fig 2C and 2D). For CT, patients in areas with the highest socioeconomic vulnerability have higher utilization than patients in areas with the lowest socioeconomic vulnerability for all BMI groups. For MR, this remains true only for the underweight BMI group, with overweight and obese cohorts all showing lower MR utilization in patients with highest SES vulnerability. Males show higher CT and MR utilization than females for cohorts with BMI less than 30, but this difference is lost at higher BMI ranges (Fig 2E and 2F). In fact, MR utilization in males with moderate or severe obesity falls below that of females of the same BMI classes (Fig 2F). To supplement the above analyses, Tables 2 and 3 show CT and MR utilization rates, respectively, for each 4-way combination of age, sex, SES vulnerability, and BMI.

To compare the CT and MR utilization data to radiology utilization in general, we computed, for the same cohorts as above, the proportion of patients with any radiology department encounter in 2021 (Fig 3). Interestingly, the disproportionately high utilization rates for CT and MR among underweight individuals is not observed when examining radiology

**Table 3. MR utilization rates for each combination of sex, age (in years), SES SVI vulnerability quartile, and BMI.**

| Sex | Age | SES SVI | <18.5 | 18.5–24.9 | 25–29.9 | 30–34.9 | 35–39.9 | 40–50 | >50 | Total |
|---|---|---|---|---|---|---|---|---|---|---|
| | | | | | | BMI | | | | |
| F | 18–39 | Q1 | 3.3% | 2.6% | 3.0% | 3.4% | 3.8% | 4.0% | 4.3% | 3.0% |
| F | 18–39 | Q2 | 3.3% | 2.6% | 3.0% | 3.4% | 3.8% | 3.9% | 4.1% | 3.1% |
| F | 18–39 | Q3 | 3.4% | 2.7% | 3.0% | 3.4% | 3.7% | 3.7% | 4.0% | 3.1% |
| F | 18–39 | Q4 | 3.2% | 2.4% | 2.7% | 2.9% | 3.1% | 3.2% | 3.3% | 2.7% |
| F | 18–39 | N/A | 2.4% | 1.9% | 2.6% | 3.0% | 3.2% | 3.0% | 3.2% | 2.4% |
| F | 40–65 | Q1 | 9.1% | 5.3% | 5.8% | 6.3% | 6.6% | 6.5% | 6.0% | 5.8% |
| F | 40–65 | Q2 | 10.1% | 6.0% | 6.2% | 6.7% | 7.0% | 6.7% | 6.2% | 6.2% |
| F | 40–65 | Q3 | 11.1% | 6.6% | 6.7% | 7.1% | 7.4% | 7.0% | 6.4% | 6.6% |
| F | 40–65 | Q4 | 11.3% | 6.8% | 6.5% | 6.7% | 6.9% | 6.6% | 5.9% | 6.3% |
| F | 40–65 | N/A | 7.5% | 4.3% | 4.6% | 5.2% | 5.1% | 5.3% | 5.6% | 4.6% |
| F | >65 | Q1 | 11.8% | 8.9% | 8.9% | 9.1% | 9.1% | 8.5% | 8.2% | 8.6% |
| F | >65 | Q2 | 11.5% | 8.9% | 8.9% | 9.0% | 9.1% | 8.4% | 7.7% | 8.5% |
| F | >65 | Q3 | 11.6% | 9.2% | 9.0% | 9.1% | 9.3% | 8.5% | 8.3% | 8.7% |
| F | >65 | Q4 | 11.7% | 9.2% | 8.8% | 8.9% | 8.9% | 8.3% | 7.6% | 8.5% |
| F | >65 | N/A | 9.5% | 7.3% | 6.8% | 6.8% | 7.1% | 6.6% | 5.6% | 6.6% |
| M | 18–39 | Q1 | 3.0% | 2.4% | 2.7% | 2.9% | 3.0% | 3.0% | 2.8% | 2.7% |
| M | 18–39 | Q2 | 3.1% | 2.4% | 2.7% | 2.9% | 3.1% | 2.9% | 2.7% | 2.7% |
| M | 18–39 | Q3 | 3.5% | 2.4% | 2.7% | 2.9% | 3.1% | 2.9% | 2.5% | 2.7% |
| M | 18–39 | Q4 | 3.5% | 2.3% | 2.5% | 2.6% | 2.6% | 2.5% | 2.2% | 2.4% |
| M | 18–39 | N/A | 2.7% | 1.8% | 2.2% | 2.3% | 2.5% | 3.0% | 3.0% | 2.2% |
| M | 40–65 | Q1 | 13.6% | 6.3% | 5.4% | 5.5% | 5.8% | 5.5% | 4.4% | 5.6% |
| M | 40–65 | Q2 | 13.6% | 7.0% | 5.9% | 5.8% | 6.0% | 5.6% | 4.5% | 5.9% |
| M | 40–65 | Q3 | 13.6% | 7.6% | 6.3% | 6.2% | 6.2% | 5.7% | 4.7% | 6.2% |
| M | 40–65 | Q4 | 13.8% | 7.5% | 6.1% | 5.9% | 6.1% | 5.4% | 4.4% | 5.9% |
| M | 40–65 | N/A | 11.2% | 5.5% | 4.6% | 4.4% | 4.9% | 4.9% | 3.9% | 4.6% |
| M | >65 | Q1 | 15.1% | 10.6% | 9.4% | 9.2% | 9.3% | 8.5% | 7.9% | 9.3% |
| M | >65 | Q2 | 14.2% | 10.4% | 9.1% | 8.9% | 8.8% | 8.2% | 7.8% | 9.0% |
| M | >65 | Q3 | 14.5% | 10.5% | 9.3% | 9.0% | 8.8% | 8.0% | 7.4% | 9.1% |
| M | >65 | Q4 | 14.3% | 10.4% | 9.0% | 8.6% | 8.4% | 7.6% | 7.7% | 8.9% |
| M | >65 | N/A | 10.4% | 7.6% | 6.7% | 6.3% | 6.5% | 5.7% | 4.7% | 6.6% |

department encounters in general, for which radiography is the most commonly performed imaging modality. The underweight cohort demonstrated a radiology department utilization rate of 34.4 ± 0.1% (p<0.0001 relative to normal BMI), slightly lower than the normal-weight cohort, which had a radiology department visit utilization of 34.7 ± 0.02%. Class 2 obese patients had the highest prevalence (41.0 ± 0.04%; (p<0.0001 relative to normal BMI) of a radiology department encounter in 2021. The association between BMI and any radiology utilization is shown in Fig 3A. Younger age, male gender, and higher SES vulnerability are associated with lower prevalence of radiology department encounters, without an apparent confounding interaction.

## 4.3 Estimating effects of BMI on morbidity and mortality

Because imaging utilization in a cohort depends in part on prevalence of high-morbidity diseases, we attempted to assess relative disease severity in each cohort by quantifying emergency department visits and mortality (Figs 3 & 4, respectively). Underweight patients and severe

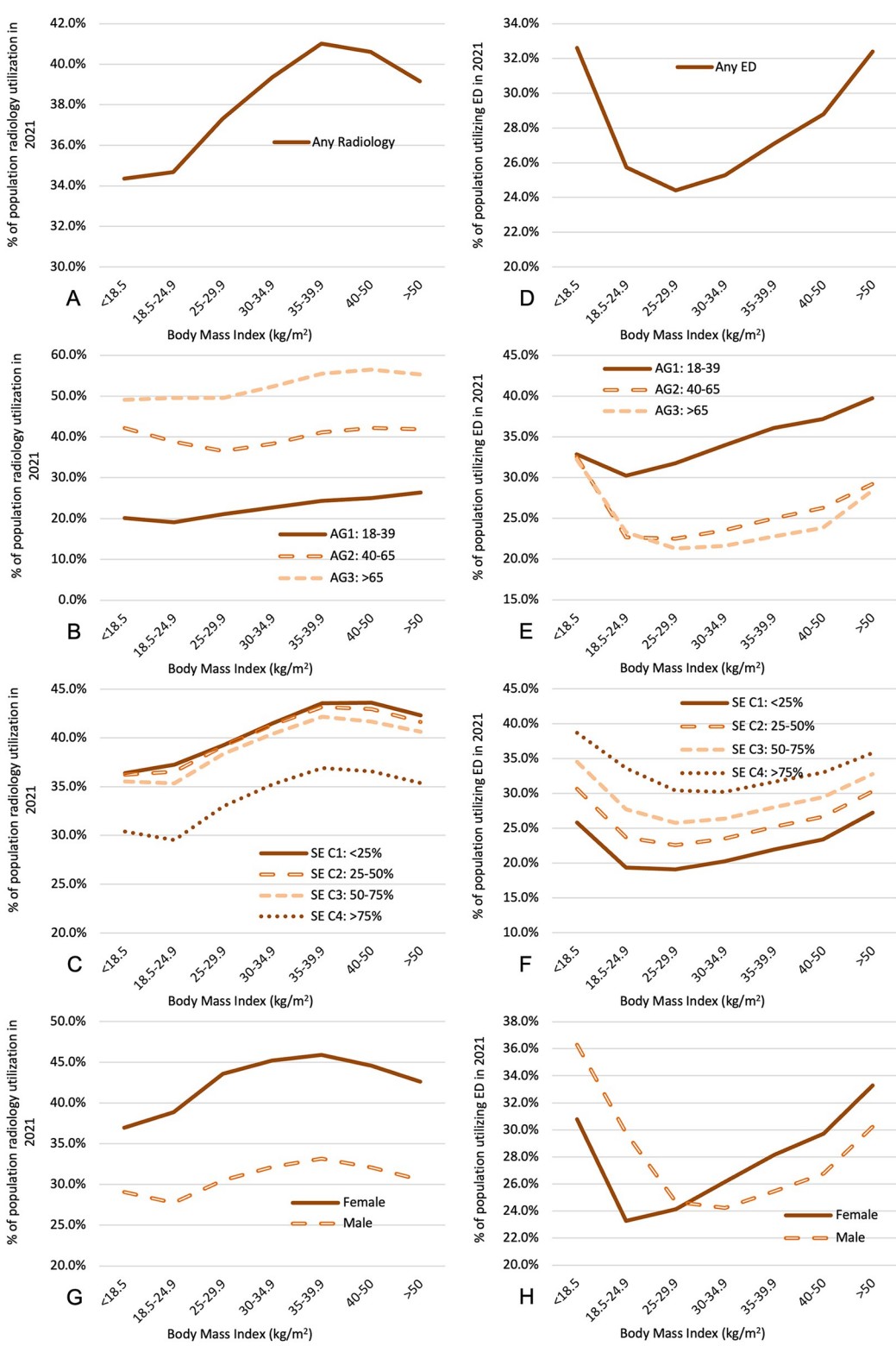

**Fig 3.** Association between BMI and the percentage of the cohort with any radiology encounter in 2021 (**A**) stratified by age (**B**), SES (**C**), and sex (**G**). Association between BMI and the percentage of the cohort with any ED encounter in 2021 (**D**) stratified by age (**E**), SES (**F**), and sex (**H**). AG = age group; SE = socioeconomic status social vulnerability category (C1 presents the lowest quartile of vulnerability).

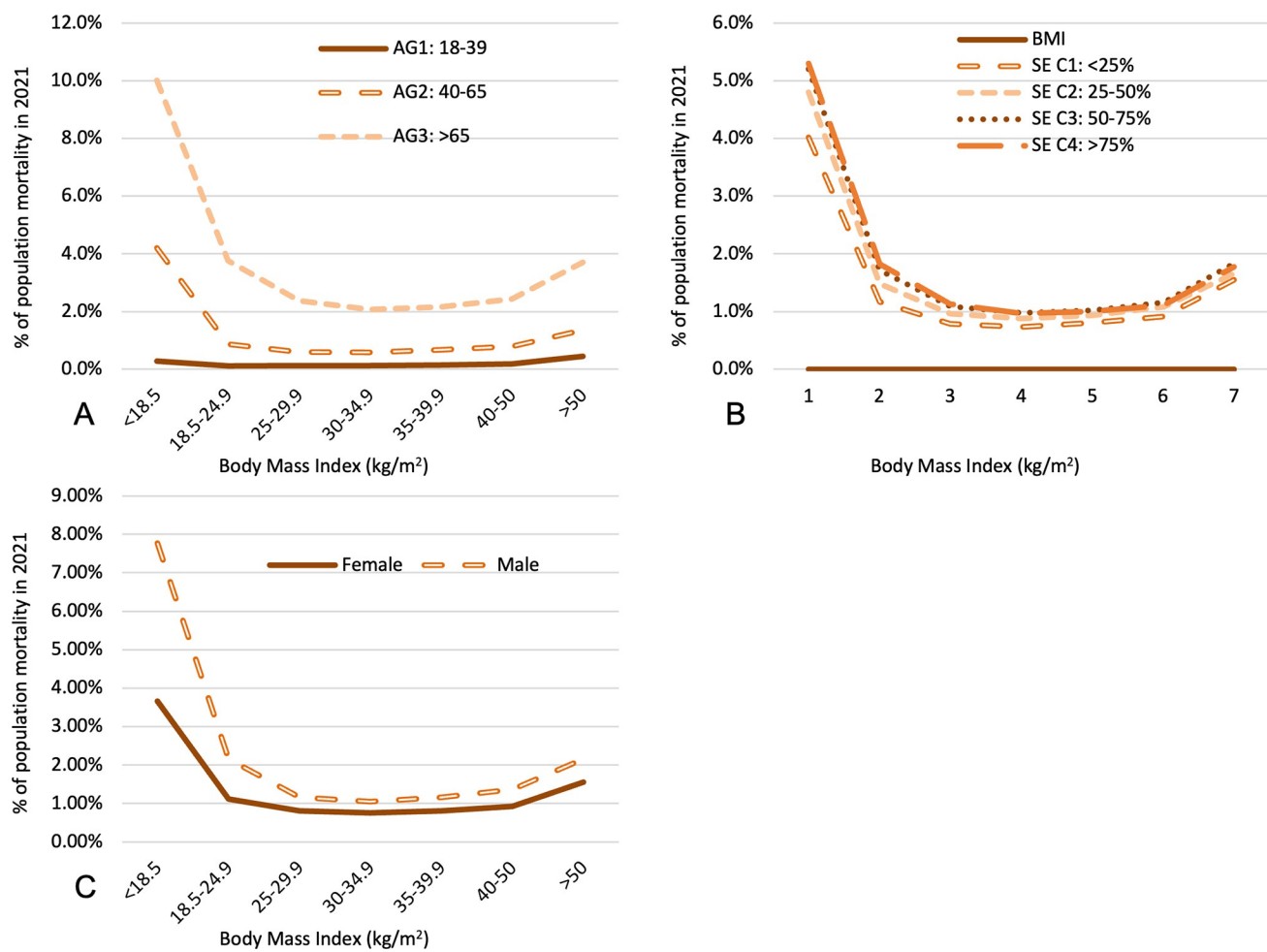

**Fig 4.** Association between BMI and population mortality in 2021 stratified by age (**A**), SES (**B**), and sex (**C**). AG = age group; SE = socioeconomic status social vulnerability category (C1 presents the lowest quartile of vulnerability).

Class 3 obese patients (BMI > 50) were found to have the highest ED utilization, at 32.6 ± 0.1% and 32.4 ± 0.1% (p<0.0001 relative to normal BMI), respectively. Overweight status had a mildly protective effect when examining ED visits, with the overweight cohort showing the lowest ED utilization at 24.4 ± 0.02% (p<0.0001 relative to normal BMI), compared to 25.7 ± 0.02% for normal-weight individuals. Interestingly, younger patients are associated with a higher rate of ED visits, with a larger age effect at higher BMI ranges. Higher SES vulnerability is associated with a higher rate of ED visits and a higher BMI range at which minimum ED utilization was observed.

A similar U-shaped dependence was observed with mortality and BMI (Fig 4). Age-adjusted mortality was highest in the underweight group (4.8 ± 0.04%) (p<0.0001 relative to normal BMI). Increasing BMI from normal to the mildly obese range decreased age-adjusted mortality from 1.5 ± 0.01% to 1.0 ±0.01% (p<0.0001). Further increases in BMI resulted in higher age-adjusted mortality, reaching 1.8 ± 0.03% for BMI > 50 (p<0.0001 relative to normal BMI). At all age groups examined, the underweight group and the severe Class 3 obese group (BMI>50) had the highest mortalities. In patients > 65 years of age, underweight patients showed a mortality of 10.0 ± 0.1% (p<0.0001 relative to normal BMI) compared to 3.7 ± 0.1% (p<0.0001 relative to normal BMI) for the severe Class 3 obese group. For patients

18–39 years, for whom mortality rates are generally much lower, the severe Class 3 obese group had higher mortality (0.44 ± 0.02%) (p<0.001 relative to normal BMI) than the underweight group (0.27 ± 0.01%, p<0.0001 relative to normal BMI). Interestingly, overweight or mildly obese status had protective effects on mortality among older patients, with the minimum mortality rates among older patients found in the class 1 obese cohort (0.59 ± 0.006% and 2.1 ± 0.01% for patients aged 40–65 years and >65 years, respectively, p<0.0001 relative to normal BMI). No significant interaction between mortality and SES is appreciated.

### 4.4 Subgroup analysis of CT and MR utilization by diagnosis groupings

Following identification of subgroups defined by common diagnoses, the quantitative relationships between same-year CT or MR utilization and BMI were examined for these diagnosis-related subgroups (Fig 5). CT utilization across BMI for most of these subgroups is similar to findings for the entire study population, except for the heart failure cohort, for which CT utilization monotonically decreases with BMI. Similarly, the dependence of MR utilization on BMI for most of the examined subgroups is similar to the entire study population, except that cerebrovascular disease shows a slight increase in MR utilization from mild to severe obesity. Of note, these subgroups are the smallest among the subgroups examined.

## 5. Discussion

In this large-scale study of approximately 41.5 million patients across multiple institutions, geographic regions, payors, and practice settings, we examined quantitative relationships between BMI and imaging utilization. For our primary outcome variable of per capita CT and MR utilization, we found that there is not a simple linear dependence between BMI and CT/ MR utilization; instead, underweight patients had disproportionately high CT and MR utilization compared to normal weight, overweight, or obese patients, and high extremes of BMI are associated with mildly higher CT and lower MR utilization than for the normal weight cohort. The underweight population and BMI >50 populations have similarly low prevalence in our study population, but the underweight patients show much greater utilization of these high-cost imaging modalities than the severe class 3 obese patients. These trends are different than for radiology department encounters in general, for which moderately obese patients have the highest utilization rates.

We explored potential explanations of these observations. As age is associated with higher CT and MR utilization, presumably due to higher prevalence of various comorbidities with advanced age that contribute to a greater need for imaging, the above findings may be disproportionately impacted by the older patients in our population, but confounding effects of age or socioeconomic status do not appear to account for our main findings of utilization rates at the high and low extremes of BMI. Our findings of low SES being associated with lower utilization of high-cost modalities of CT and MR are corroborated by previous investigations on other facets of healthcare utilization [14, 15]. Because imaging utilization likely depends on numerous variables, including prevalence of associated disease processes, we examined mortality rates and ED utilization rates as metrics for severity of comorbidities, and for both these outcome variables, we documented a U-shaped relationship, in which underweight and severely obese subpopulations experience increased risk of mortality, regardless of age or SES, corroborating existing literature on BMI and mortality [16–21]. In fact, the overweight cohort was found to have lower ED utilization than normal-weight patients in the study population. This could be due to limitations of BMI as a measurement tool, such as abundance of high muscle mass participants in the overweight BMI cohort [22], but could also reflect a protective benefit of being moderately overweight as discussed previously [17, 23].

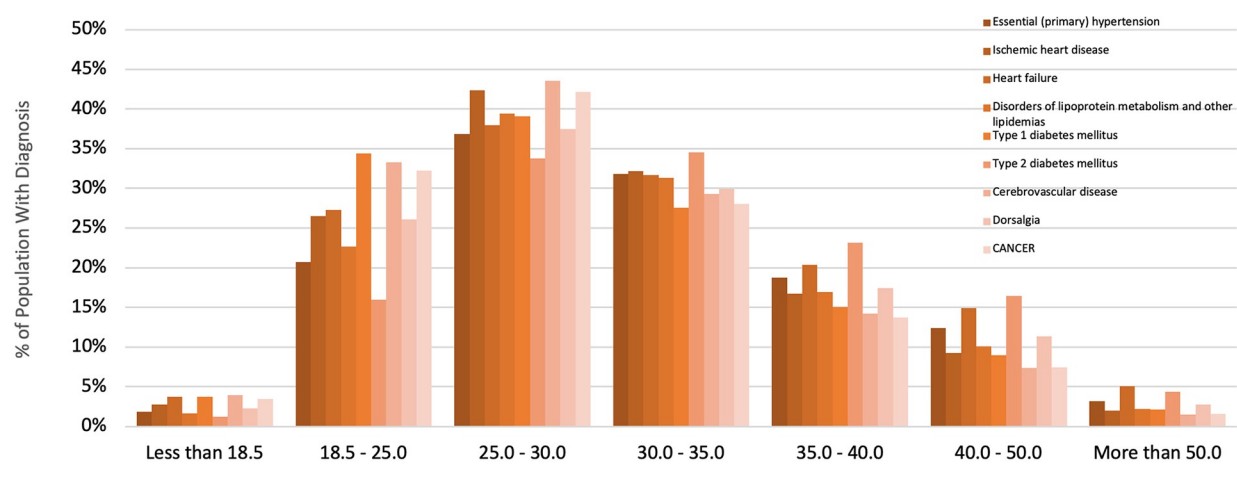

**Fig 5.** Histogram of patients with selected diagnoses across BMI cohorts (**A**) and the percentage of each BMI cohort undergoing CT (**B**) and MR (**C**) medical imaging stratified by diagnosis.

If imaging utilization were proportional to underlying disease morbidity, it would be logical to predict a similar U-shaped relationship between BMI and imaging utilization. Indeed, regardless of age or SES, the underweight cohort was found to have the highest CT and MR imaging utilization. Similarly, the lowest mortality and ED utilization rates were found in normal to mildly obese patients, depending on age and SES, and these patients also show the lowest CT utilization rates. Severely obese patients, not unexpectedly, have higher rates of ED utilization, higher mortality, and higher CT utilization, but they show lower MR utilization, suggesting a mismatch between the potentially higher clinical need for MR imaging in this higher-morbidity and higher-mortality group and the observed lower utilization of MR. The quantitative relationship between CT/MR utilization and BMI persists across multiple diagnosis-related subgroups, with only a few exceptions related to the less prevalent diagnoses examined, suggesting that the observed overall variation across BMI cannot simply be explained by confounding effects from a few highly represented diagnoses.

We acknowledge that relationships between BMI and imaging utilization are complex. Certain disease processes may be over-represented or under-represented among high-BMI cohorts relative to normal-BMI cohorts, and each disease process may be associated with a likelihood of imaging as part of workup or surveillance. Even when considering a single diagnosis, a higher BMI may place patients at higher or lower risk of complications than others of the same diagnosis. These complex interactions render it challenging to define a simple quantitative model to describe the BMI-imaging utilization relationships, and it is difficult to identify true causal relationships. Nonetheless, we propose a few plausible explanations for the observed lower rates of MR imaging in moderate to severe obesity. One explanation is limited availability of MR systems able to accommodate large patients due to limited bore size, patient claustrophobia or other intolerance, table weight limits, and potentially longer image acquisition times [24]. Body habitus may also affect MR image quality, such that increased noise and artifacts, image cropping, and limited field of view may render acquired images less diagnostic [11, 25–27]. It is possible that demand for some MR exams, such as those used for surgical planning, may be less in severely obese patients due to decreased surgical candidacy related to obesity or related comorbidities. Other factors, biases among healthcare providers towards obese populations or reluctance of obese patients to seek care due to stigmatization or other access limitations could also represent barriers to MR imaging of obese patients [28, 29]. To our knowledge, the quantitative association between BMI and imaging utilization has not been formally addressed in a large multi-institutional, multi-payor population in prior published investigations.

This study offers several clinical implications. Considering the disproportionately elevated mortality and imaging utilization rates among underweight populations, it is apparent that underweight status does equate to being healthy but rather poses substantially higher mortality risk and greater high-cost imaging utilization than overweight or mildly obese individuals. Although this study was not designed to identify the causes of higher mortality or imaging utilization at the high and low BMI extremes, we have observed this pattern across multiple diagnosis-related subgroups. Additionally, the overall elevated mortality and ER utilization associated with severe obesity are in contrast to the decreased MR imaging utilization seen in this study. While further investigations are warranted into the potential causes of this apparent mismatch between higher morbidity and lower MR utilization among severely obese patients, efforts related to improving access to or technical feasibility of MRI for obese patients may be helpful for optimal care of this high-risk population.

Our study has several limitations. The study population, while large, may not necessarily represent the general population, since our inclusion criteria requires a recorded BMI in 2021 at a healthcare institution participating in Cosmos; therefore, there is a selection bias favoring

patients with recent or more frequent access to healthcare. Additionally, the informatics platform used in our study permits only acquisition of aggregate counts rather than patient-level data elements, precluding use of regression or more sophisticated predictive modeling approaches.

In conclusion, population utilization of CT and MR varies with BMI, but the dependence is complex. In our large, geographically diverse, multi-institutional, multi-payor patient population, both the high and low extremes of BMI are associated with greater morbidity and mortality metrics and higher CT imaging utilization, but MR utilization is actually lower among obese populations than normal-weight patients. Additional studies are warranted to elucidate the underlying causes of these observations. This research may pave the way for more advanced predictive modeling approaches to investigate the effects of comorbidities and other risk factors on imaging utilization rates.

## Supporting information

**S1 File. BMI data tables.** A set of data tables containing counts of various subsets of the study population.
(XLSX)

## Author Contributions

**Conceptualization:** Mina Dawod, Paul Nagib, John Zaki, Luciano M. Prevedello, Amna A. Ajam, Xuan V. Nguyen.

**Data curation:** Mina Dawod, Paul Nagib, John Zaki, Luciano M. Prevedello, Amna A. Ajam, Xuan V. Nguyen.

**Formal analysis:** Mina Dawod, John Zaki, Luciano M. Prevedello, Amna A. Ajam, Xuan V. Nguyen.

**Investigation:** Mina Dawod, Paul Nagib, Xuan V. Nguyen.

**Methodology:** Mina Dawod, Paul Nagib, John Zaki, Xuan V. Nguyen.

**Project administration:** Xuan V. Nguyen.

**Supervision:** Xuan V. Nguyen.

**Validation:** Mina Dawod, Xuan V. Nguyen.

**Visualization:** Mina Dawod, Xuan V. Nguyen.

**Writing – original draft:** Mina Dawod, Paul Nagib, John Zaki, Luciano M. Prevedello, Amna A. Ajam, Xuan V. Nguyen.

**Writing – review & editing:** Mina Dawod, Paul Nagib, John Zaki, Luciano M. Prevedello, Amna A. Ajam, Xuan V. Nguyen.

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
