## [Decision Letter · Decision Letter 0]

10 Mar 2024

PONE-D-23-43698CT and MR Utilization and Morbidity Metrics Across Body Mass IndexPLOS ONE

Dear Dr. Nguyen,

Thank you for submitting your manuscript to PLOS ONE. After careful consideration, we feel that it has merit but does not fully meet PLOS ONE’s publication criteria as it currently stands. Therefore, we invite you to submit a revised version of the manuscript that addresses the points raised during the review process.

We look forward to receiving your revised manuscript.

Kind regards,

Amir Hossein Behnoush

Academic Editor

PLOS ONE

Journal Requirements:

2. For studies involving third-party data, we encourage authors to share any data specific to their analyses that they can legally distribute. PLOS recognizes, however, that authors may be using third-party data they do not have the rights to share. When third-party data cannot be publicly shared, authors must provide all information necessary for interested researchers to apply to gain access to the data. (https://journals.plos.org/plosone/s/data-availability#loc-acceptable-data-access-restrictions) 

3. Please ensure that you include a title page within your main document. You should list all authors and all affiliations as per our author instructions and clearly indicate the corresponding author.

5. Please include your tables as part of your main manuscript and remove the individual files. Please note that supplementary tables (should remain/ be uploaded) as separate ""supporting information"" files

Reviewers' comments:

Reviewer's Responses to Questions

**Comments to the Author**

1. Is the manuscript technically sound, and do the data support the conclusions?

Reviewer #1: Yes

Reviewer #2: Yes

2. Has the statistical analysis been performed appropriately and rigorously? 

Reviewer #1: Yes

Reviewer #2: Yes

3. Have the authors made all data underlying the findings in their manuscript fully available?

Reviewer #1: Yes

Reviewer #2: Yes

4. Is the manuscript presented in an intelligible fashion and written in standard English?

Reviewer #1: Yes

Reviewer #2: Yes

5. Review Comments to the Author

Reviewer #1: The study titled "CT and MR Utilization and Morbidity Metrics Across Body Mass Index" investigated the association between BMI groups and CT and MRI utilization. The study is well-written. Strengths of this study include large sample size and comprehensive analysis. However, comorbidities have not been considered as a covariable. I have some comments for improvement:

- Abbreviations should be defined in their first use. I found several abbreviations that have not been defined in their first use. Make sure you are only using abbreviated forms after the definition.

- Add more data to the knowledge gap of the introduction.

- Upload figures with higher quality in the revised version.

Reviewer #2: This is an interesting study where the authors have collected a very large and unique dataset. The paper is generally well written and structured. However, in my opinion the paper has some shortcomings in regard to some data analyses and text, and I feel this dataset has not been utilized to its full extent. I believe that the section 3, data analysis, needs some more words regarding the statistical analysis. Furthermore, it would be great if we also know the sex, as in older ages, women use imaging utilization more often as a consequence of osteoporosis and falls. I think that there is a deeper explanation regarding BMI and imaging utilization, and it would be nice if the authors explain the reasons beyond the BMI. Moreover, the authors haven’t explained the SES groups, which socioeconomic class has the lowest income? Also, I suggest the authors to cite more literature.

6. PLOS authors have the option to publish the peer review history of their article (what does this mean?). If published, this will include your full peer review and any attached files.

Reviewer #1: No

Reviewer #2: No

---

## [Author Response · Author response to Decision Letter 0]

2 May 2024

Response to Reviewers (please refer to the uploaded Response to Reviewers document)

We would like to thank the reviewers for their insightful comments and constructive feedback. The manuscript has been revised to address all reviewer comments, and we believe it has improved substantially in clarity and rigor. Based on some of the reviewer suggestions to evaluate other variables, we re-queried the database to obtain additional data elements. Keep in mind that because the Cosmos system is updated periodically as new institutions are added, any new data query would produce slightly different numbers. We have meticulously updated all results text, tables, and figures to reflect the data from the most recent data query, which now includes 49 million patients, which is more than the 41.5 million patients available at the time of our initial data query. Responses to individual comments are listed below.

Reviewer #1: The study titled "CT and MR Utilization and Morbidity Metrics Across Body Mass Index" investigated the association between BMI groups and CT and MRI utilization. The study is well-written. Strengths of this study include large sample size and comprehensive analysis. However, comorbidities have not been considered as a covariable. I have some comments for improvement:

- Abbreviations should be defined in their first use. I found several abbreviations that have not been defined in their first use. Make sure you are only using abbreviated forms after the definition.

We have updated the manuscript to ensure that all abbreviations are spelled out at first mention and that only abbreviated terms are used subsequently.

- Add more data to the knowledge gap of the introduction.

The introduction has been updated with a few additional references and statements to more clearly describe the knowledge gap.

- Upload figures with higher quality in the revised version.

Figures have been processed through the PACE software to comply with the journal’s image quality standards.

Reviewer #2: This is an interesting study where the authors have collected a very large and unique dataset. The paper is generally well written and structured. However, in my opinion the paper has some shortcomings in regard to some data analyses and text, and I feel this dataset has not been utilized to its full extent. I believe that the section 3, data analysis, needs some more words regarding the statistical analysis. Furthermore, it would be great if we also know the sex, as in older ages, women use imaging utilization more often as a consequence of osteoporosis and falls. 

We have requeried the database to obtain patient sex as an additional independent variable. Note that because the database is periodically updated, any requery results in minor changes to the previously reported numbers. We have updated all figures and tables and have ensured all references to quantitative data in the text remain accurate based on the most recent data query. 

To allow greater transparency into the effects of various combinations of variables (such as older females) on CT and MR utilization, Tables 2 and 3 have been added to list utilization metrics for each 4-way combination of age, sex, SES SVI quartile, and BMI. 

Additional details of statistical analyses have been added. Specifically, inferential statistics have been added to document statistical significance of each reported proportion relative to the corresponding normal-BMI cohort. P values related to statistical significance have been added to the text of Results.

I think that there is a deeper explanation regarding BMI and imaging utilization, and it would be nice if the authors explain the reasons beyond the BMI. 

We acknowledge that relationships between BMI and imaging utilization are complex. Certain disease processes may be over-represented or under-represented among high-BMI cohorts relative to normal-BMI cohorts, and each disease process may be associated with a likelihood of imaging as part of workup or surveillance. Even when considering a single diagnosis, a higher BMI may place patients at higher or lower risk of complications than others of the same diagnosis. These complex interactions render it challenging to define a simple quantitative model to describe the BMI-imaging utilization relationships, and it is difficult to identify true causal relationships. 

We have incorporated these additional statements into the discussion.

Moreover, the authors haven’t explained the SES groups, which socioeconomic class has the lowest income? Also, I suggest the authors to cite more literature.

We have revised the Methods section to more clearly describe the SES component of the Social vulnerability index. In addition, more literature has been cited in the discussion.

---

## [Decision Letter · Decision Letter 1]

11 Jun 2024

CT and MR Utilization and Morbidity Metrics Across Body Mass Index

PONE-D-23-43698R1

Dear Dr. Nguyen,

We’re pleased to inform you that your manuscript has been judged scientifically suitable for publication and will be formally accepted for publication once it meets all outstanding technical requirements.

Kind regards,

Amir Hossein Behnoush

Academic Editor

PLOS ONE

Additional Editor Comments (optional):

Reviewers' comments:

Reviewer's Responses to Questions

**Comments to the Author**

1. If the authors have adequately addressed your comments raised in a previous round of review and you feel that this manuscript is now acceptable for publication, you may indicate that here to bypass the “Comments to the Author” section, enter your conflict of interest statement in the “Confidential to Editor” section, and submit your "Accept" recommendation.

Reviewer #1: All comments have been addressed

2. Is the manuscript technically sound, and do the data support the conclusions?

Reviewer #1: (No Response)

3. Has the statistical analysis been performed appropriately and rigorously? 

Reviewer #1: (No Response)

4. Have the authors made all data underlying the findings in their manuscript fully available?

Reviewer #1: (No Response)

5. Is the manuscript presented in an intelligible fashion and written in standard English?

Reviewer #1: (No Response)

6. Review Comments to the Author

Reviewer #1: (No Response)

7. PLOS authors have the option to publish the peer review history of their article (what does this mean?). If published, this will include your full peer review and any attached files.

Reviewer #1: No

---

## [Editor Report · Acceptance letter]

19 Jun 2024

PONE-D-23-43698R1 

PLOS ONE

Dear Dr. Nguyen, 

I'm pleased to inform you that your manuscript has been deemed suitable for publication in PLOS ONE. Congratulations! Your manuscript is now being handed over to our production team.

Kind regards, 

on behalf of

Dr. Amir Hossein Behnoush 

Academic Editor

PLOS ONE